# Identification and Validation of Hub Genes in the Stenosis of Arteriovenous Fistula

**DOI:** 10.3390/jpm13020207

**Published:** 2023-01-25

**Authors:** Yu Li, Yue Chen, Wenhao Cui, Jukun Wang, Xin Chen, Chao Zhang, Linzhong Zhu, Chunjing Bian, Tao Luo

**Affiliations:** Department of General Surgery, Xuanwu Hospital, Capital Medical University, Beijing 100053, China

**Keywords:** arteriovenous fistula, bioinformatic analysis, hub gene, *FOS*

## Abstract

Arteriovenous fistula (AVF) is the most widely used hemodialysis vascular access in China. However, stenosis of the AVF limits its use. The mechanism of AVF stenosis is currently unknown. Therefore, the purpose of our study was to explore the mechanisms of AVF stenosis. In this study, we identified the differentially expressed genes (DEGs) based on the Gene Expression Omnibus (GEO) dataset (GSE39488) between venous segments of AVF and normal veins. A protein–protein interaction (PPI) network was constructed to identify hub genes of AVF stenosis. Finally, six hub genes (*FOS*, *NR4A2*, *EGR2*, *CXCR4*, *ATF3*, and *SERPINE1*) were found. Combined with the results of the PPI network analysis and literature search, *FOS* and *NR4A2* were selected as the target genes for further investigation. We validated the bioinformatic results via reverse transcription PCR (RT-PCR) and Western blot analyses on human and rat samples. The expression levels of the mRNA and protein of *FOS* and *NR4A2* were upregulated in both human and rat samples. In summary, we found that FOS may play an important role in AVF stenosis, which could be a potential therapeutic target of AVF stenosis.

## 1. Introduction

Chronic kidney disease (CKD) is a condition causing irreversible destruction of renal parenchyma, with a progressive loss of kidney function over several years. Maintenance hemodialysis (MHD) is the best alternative to renal transplantation due to the shortage of human donor organs. Patients undergoing MHD need patent vascular access for optimal treatment, and autologous arteriovenous fistula (AVF) is the optimal vascular access. However, AVF stenosis, which could be attributed to a variety of factors, may limit its use. Our previous study demonstrated that hypercholesterolemia and hyperphosphatemia are independent risk factors for AVF stenosis in patients with MHD [1]. Meanwhile, many factors caused by renal failure, such as high blood calcium, phosphorus and lipids, C-reactive protein (CRP), and serum creatinine (Scr), may affect the patency of AVF [2,3,4]. Since the mechanism of AVF stenosis is currently unclear, the purpose of our study was to explore the mechanism of AVF stenosis through bioinformatic analysis and experimental validation.

Bioinformatic analysis has been used to identify novel biomarkers of many different diseases [5,6]. Moreover, the rapid development of high-throughput technology has made research on disease-related biomarkers more feasible and reliable [7]. This has given researchers the capacity to detect all genes within several samples to screen differentially expressed genes (DEGs). Therefore, bioinformatics could improve the understanding of AVF stenosis. The major online genetic database involving vascular diseases is the Gene Expression Omnibus (GEO) (https://www.ncbi.nlm.nih.gov/geo/, accessed on 1 July 2022). In addition, online tools, such as GEO2R (https://www.ncbi.nlm.nih.gov/geo/geo2r/, accessed on 1 July 2022), Network Analyst (https://www.networkanalyst.ca/, accessed on 1 July 2022), and the Search Tool for the Retrieval of Interacting Genes database (STRING) (https://cn.string-db.org/, accessed on 1 July 2022), have been widely used. All of these bioinformatic tools are useful for identifying hub genes related to AVF stenosis. Although some hub genes have been reported previously [8], these studies may not be enough.

In our study, GSE39488 was downloaded to screen DEGs between the control and AVF group. Thereafter, a protein–protein interaction (PPI) network of DEGs was constructed; hub genes with a higher degree were identified. All processes in this study were performed using RStudio and Cytoscape software and related online tools.

## 2. Bioinformatics Analysis

### 2.1. Microarray Data

The GSE39488 dataset, which contains the gene expression profile data of AVF patients, was obtained from the GEO database [9]. This dataset includes six AVF tissues and four normal veins. Sequencing data were acquired using the GPL10332 platform (Agilent-026652 Whole Human Genome Microarray 4 × 44K v2).

### 2.2. Identification of the DEGs

The raw data of gene expression profiles from GEO were preprocessed for background correction (duplicate removal, nonsense gene removal, etc.) and data normalization (log2 transformation) using RStudio software version 1.1.383. The DEGs between AVF tissues and normal veins were identified via Network Analyst online tool. The genes with a log2 fold-change (FC) of >1 and an adjusted *p*-value of <0.05 were considered to be differentially expressed.

### 2.3. Construction of the PPI Network and Identification of Hub Genes

STRING online database was used to construct the PPI network based on the DEGs which were previously identified. Cytoscape software version 3.7.2 was used to visualize the PPI network [10]. Thereafter, we calculated the degree of all genes. The genes with higher degree values were considered hub genes.

## 3. Validation of Bioinformatic Results

### 3.1. Patients and Sample Collection

A total of five patients who required AVF repair and three patients who required AVF creation were recruited from the Department of General Surgery, Xuanwu Hospital. The clinical characteristics of these patients, including age, gender, smoking, hypertension, and diabetes mellitus, were collected and compared between groups. The results presented in Table 1 show that all of these clinical characteristics were not statistically different between the two groups. A 0.5 cm segment of vein from stenotic AVF was collected as the test group during the operation of AVF repair. A similarly sized vein segment collected during AVF creation was used as the negative control (NC). Each sample was equally divided into two parts for reverse transcription (RT-PCR) and Western blot analysis, and these samples were stored in liquid nitrogen until further use.

### 3.2. Animal Model Creation and Sample collection

Our present study involved 20 male Sprague–Dawley rats (Vital River Laboratory Animal Technology Co., Ltd., Beijing, China) aged 6 months. The AVFs were created in an end-to-side manner between the right external jugular vein and the right common carotid artery. The animals were euthanized 7, 14, or 28 days after the surgical procedure. The normal right external jugular vein served as the control group. Heparin (KEHBIO, Beijing, China), dosed at 0.1 IU/g, was used to prevent acute AVF thrombosis after the operation.

Isoflurane inhalation anesthesia with oxygen-enriched (40%) air was used, with isoflurane concentrations of 4% for induction and 1.5% for maintenance. After anesthesia, the animals were fixed supine on a heating blanket, and the eyes were covered with a piece of gauze. Thereafter, the surgical area was shaved and disinfected. Finally, a median neck incision was made under a YZ20T4 dissecting microscope (66Vision·Tech, Suzhou, China).

The surgical procedures were as follows: Firstly, the right common carotid artery was dissected carefully. After, two vascular clamps were placed proximally and distally. A 5 mm incision was made longitudinally on the wall of the common carotid artery. Meanwhile, we dissected the right sternocleidomastoid muscle and transected it from the middle. Next, the right external jugular vein was detected and clamped proximally and ligatured distally, followed by a transection from the distal. Thereafter, the artery and vein were rinsed with saline containing 100 IU/mL heparin, whereupon the end of the vein was anastomosed to the side of the artery with 11-0 running sutures. At the end of the process, heparin was injected intravenously into the external jugular vein at a dose of 0.1 IU/g body weight. After completion of the anastomosis, venous and arterial clamps were removed carefully, starting with the venous clamp. If bleeding occurred at the site of the anastomosis, gentle compression to stop bleeding was performed; otherwise, an additional suture was inserted. In the final step, after confirmation of the patency of the AVF, the skin was closed with a 4-0 running suture. The surgical procedure is shown in Figure 1.

Thereafter, the abdominal cavity was accessed through a midline incision from the xiphoid process to the symphysis pubis. Retractors were placed to maintain exposure. The abdominal content was reflected to the right to expose the inferior vena cava, aorta, and right kidney. The intestines were wrapped in saline-soaked gauze to keep them moist and warm. The right ureter was then carefully isolated and ligated with 8-0 prolene suture. Finally, the skin was closed with a 4-0 running suture.

Four rats were sacrificed at each time point of 7, 14, and 28 days after surgery, and the proximal external jugular vein of the AVF was resected for subsequent analysis. Each sample was equally divided into three parts, which were used for RT-PCR, Western blot, and histological analysis.

Intimal hyperplasia of the right external jugular vein was observed under an optical microscope. The area of hyperplastic intima was calculated according to the following formula: *S* (%) = $\frac{S_{2}-S_{3}}{S_{1}-S_{3}}$. *S*_1_ represents the area of the whole vein; *S*_2_ represents the area of hyperplasia intima plus residual lumen; *S*_3_ represents the area of the residual lumen. *S*_1_ and *S*_2_ were distinguished by staining. Image J software was used to calculate *S*_1_, *S*_2_, and *S*_3_ (Figure 2).

### 3.3. RT-PCR

The total RNA was extracted from the samples with Trizol reagent (Invitrogen). The RNA was then reverse-transcribed into cDNA using a Reverse Transcription Kit (Biosharp, Hefei, China). The cDNA was used as a template for SYBR Green qPCR Mix (Biosharp, Hefei, China), and RT-PCR reaction was performed using StepOnePlus Real-Time PCR System. GAPDH was used as the reference for mRNA analysis. The 2^−ΔΔCt^ method was used to calculate the expression level of mRNA of the target genes. All primers are listed in Table 2.

### 3.4. Western Blot

Proteins were extracted from the samples via radioimmunoprecipitation assay buffer (RIPA buffer; KeyGen, Nanjing, China). The protein lysates of 30 μg were loaded onto a 10% sodium dodecyl sulfate-polyacrylamide gel for electrophoresis and then transferred to a polyvinylidene fluoride (PVDF) membrane. The membrane was blocked with 5% skimmed milk powder for 0.5 h. PVDF membranes were then probed with mouse anti-FOS antibody (1:2000, ProteinTech Group, Chicago, IL, USA, 66590-1-Ig), anti-NR4A2 antibody (1:1000, ProteinTech Group, Chicago, IL, USA, 66878-1-Ig), and anti-β-actin monoclonal antibody (1:2000, ZSGB-BIO, Beijing, China, TA-09) overnight at 4 ℃. The PVDF membranes were washed with Tris-buffered saline containing Tween 20 (TBST) for 10 min × 3 and labeled with horseradish peroxidase (HRP)-conjugated secondary antibodies (1:5000; ZSGB-BIO, ZB-2305) for 1 h at RT. Immunoreactivities were detected by enhanced chemiluminescence (Biosharp, Hefei, China). The protein content was calculated by densitometry using Image J software.

### 3.5. Statistical Analysis

Statistical analyses of three independent experiments are presented as the mean ± SEM. The significance of the differences was analyzed by a one-way ANOVA test, and *p* < 0.05 was considered statistically significant. GraphPad Prism software v8.0.2 was used for the visualization of the results.

## 4. Results

### 4.1. Confirmation of the GEO Datasets

Log2 transformation was performed to normalize the raw data in the GSE39488 dataset. The results of data normalization are presented in Figure 3A. The results suggest that the mean values were approximately the same between different samples. To further validate the dataset, a principal component analysis (PCA) was conducted to test the potential association between AVF and control groups. The results of PCA suggest that the repeatability between different groups was acceptable (Figure 3B).

### 4.2. Identification of DEGs

A total of 62 DEGs were detected based on the criteria of an adjusted *p*-value of <0.05 and a log FC of >1, with 47 upregulated and 15 downregulated genes. Both a heatmap (Figure 4A) and a volcano plot (Figure 4B) were used to illustrate the DEGs.

### 4.3. Construction of PPI Network and Identification of Hub Genes

The PPI network of 62 DEGs were constructed using the STRING database, followed by Cytoscape software for further modular analysis. Finally, six genes with a higher degree were considered hub genes (Figure 5). The selected hub genes were *NR4A2*, *FOS*, *CXCR4*, *EGR2*, *SERPINE1*, and *ATF3*. *NR4A2* and *FOS* may play an important role in AVF stenosis among the hub genes. Therefore, we focused on the expression level of *FOS* and *NR4A2* in subsequent analyses.

### 4.4. Characteristics of the Patients and the Expression Levels of FOS and NR4A2

To verify the results of bioinformatics analysis, RT-PCR and Western blot analysis were performed. We collected the veins from five patients with AVF stenosis (test). However, because of the nature of the patients, we only collected three normal veins as the control (NC). As shown in Figure 6A,B, the mRNA and protein expressions of *FOS* and *NR4A2* were significantly upregulated in stenotic AVF compared to the control group, suggesting a potential relationship between relative hub genes (i.e., *NR4A2* and *FOS*) and the occurrence of AVF stenosis.

### 4.5. Expression Level of the Target Genes Increased with the Increase of Stenosis Rate

We selected 20 rats to establish an animal model to further validate the relationship between AVF stenosis and the expression of *FOS* and *NR4A2*. The surgical process is described in detail in Section 3.2. Three rats died due to an anesthetic overdose and another one due to postoperative infection. Thereafter, we sacrificed four rats and obtained samples at 7, 14, and 28 days after the operation. Meanwhile, another four rats which did not undergo surgery served as the control (NC) group. The histological analyses showed that the degree of AVF stenosis increased gradually with the increase in time after AVF creation (Figure 7). Thereafter, the results of RT-PCR and Western blot also suggest similar results, showing that both the mRNA and protein expression levels of *FOS* and *NR4A2* increased with time after AVF creation (Figure 8A,B).

## 5. Discussion

Over the past several decades, there has been a growing consensus on the “fistula first” principle in patients with MHD. The guidelines, published in 1997 and revised in 2003 and 2006 [11,12,13], suggest that AVF is the optimal choice. However, the benefits of AVF may decrease because an increasing number of elderly patients are receiving hemodialysis via AVF [14]. Some studies have suggested that altered shear stress brought by high blood flow in AVF may be responsible for AVF stenosis [15,16]. Therefore, we considered that comparing the differences between normal veins and veins exposed to high blood flow is helpful for understanding the mechanism of AVF stenosis.

To identify more specific and effective biomarkers in AVF stenosis, a combination of analyses, including bioinformatic, RT-PCR, and Western blot, based on the number of human and animal samples, was conducted. In our study, a GEO dataset (GSE39488), which included six outflow veins of AVF and four normal veins, was analyzed, and DEGs between the AVF and normal veins were identified using RStudio software. Next, a PPI network of DEGs was constructed. Finally, six hub genes (*FOS*, *NR4A2*, *EGR2*, *CXCR4*, *ATF3*, and *SERPINE1*) were identified using the STRING online tool and proceeded by Cytoscape software.

At the same time, we found that *FOS*, which is involved in the MAPK pathway, was one of the hub genes identified by our study. *FOS* can encode leucine zipper proteins, which can dimerize with proteins of the JUN family, thereby forming the transcription factor complex AP-1. As such, *FOS* have been implicated as regulators of cell proliferation, differentiation, and transformation. In some cases, the expression of *FOS* has also been associated with apoptotic cell death [17]. Jun Xu et al. have reported that the p38 MAPK pathway could be upregulated by *FOS* in vascular endothelium, which promotes the injury of endothelial cells and leads to vascular stenosis [18]. Zheng Jin et al. demonstrated that the upregulation of the FOS/MAPK signaling pathway could promote M1 macrophage polarization and suppress M2 macrophage polarization, which was one of the reasons for vascular stenosis [19]. 

Meanwhile, some studies suggested that foam cell formation may also depend on the activation of JNK2 and the p38α MAPK signaling pathways [20]. However, most of the studies were focused on the mechanisms of atherosclerosis, and studies on AVF stenosis are insufficient. For the above reasons, we evaluated the function of FOS protein when AVF stenosis occurred. RT-PCR and Western blot methods were used to validate the expression level of *FOS*. In summary, we found that the expression level of *FOS* was upregulated in stenotic AVF, which is consistent with the bioinformatics results. Moreover, the expression level will upregulate with the increase in the degree of stenosis. This showed that FOS could be a therapeutic target for AVF stenosis.

The mechanism by which *FOS* leads to AVF stenosis is complex. The results from several studies indicate that *FOS* could activate the MAPK signaling pathway [19,21]. The MAPKs, which include the extracellular signal-regulated kinases 1 and 2 (ERK-1/2), p38MAPK, and the c-jun N-terminal kinases (JNKs), are a family of highly conserved enzymes that relay extracellular signals from the cell’s cytoplasm to the nucleus [22]. The pathway could be activated by diverse stimuli, ranging from cytokines, growth factors, neurotransmitters, hormones, cellular stress, and cell adherence molecules [23]. Some studies demonstrated that the activation of the MAPK signaling pathway might cause intimal hyperplasia and the proliferation of vascular smooth muscle cells [24]. Therefore, we hold that *FOS* may cause AVF stenosis by the activation of the MAPK signaling pathway.

## 6. Conclusions

The present study identified a total of 62 DEGs between stenotic and normal veins. We identified six hub genes: *FOS*, *NR4A2*, *CXCR4*, *EGR2*, *SERPINE1*, and *ATF3*. *FOS*, which was associated with the activation of the MAPK pathway, may be responsible for AVF stenosis.

## 7. Limitations

There are still several limitations of our study. Firstly, the sample size of our study may be insufficient, and studies with a larger sample size may be necessary to confirm our results. Secondly, GSE39488 is performed by microarray profiling which may need to be updated. Finally, due to our limited resources, no in-depth study was performed. Future studies may be needed to clarify their relationship.

## Figures and Tables

**Figure 1 jpm-13-00207-f001:**
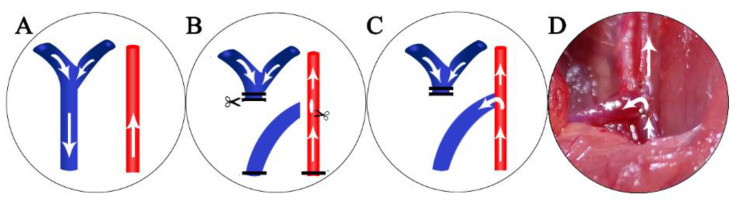
Surgical procedure: (**A**) dissection of the common carotid artery (red) and external jugular vein (blue); (**B**) after the placement of vascular clamps and ligation of the vein, an incision was made on the wall of the artery, followed by the transection of the vein; (**C**) end-to-side anastomosis was performed with running suture; (**D**) photograph of a completed AVF. The arrows indicate the direction of blood flow. AVF: autologous arteriovenous fistula.

**Figure 2 jpm-13-00207-f002:**
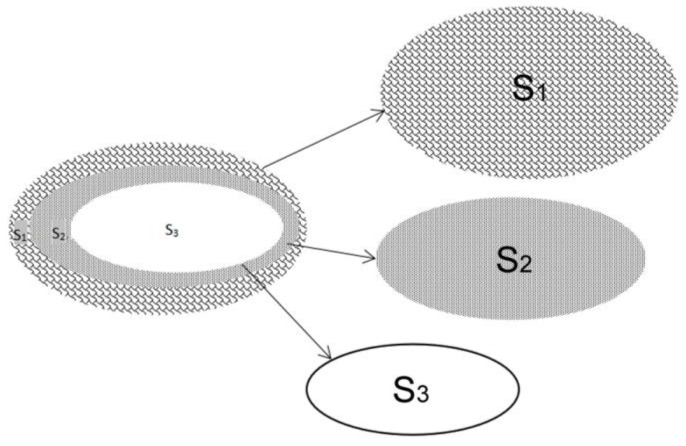
Calculation of the hyperplastic intima. *S*_1_ represents the area of the whole vein; *S*_2_ represents the area of hyperplasia intima plus residual lumen; *S*_3_ represents the area of the residual lumen. Hyperplastic endothelium was calculated on the original figures according to the following formula: *S* (%) = $\frac{S_{2}-S_{3}}{S_{1}-S_{3}}$.

**Figure 3 jpm-13-00207-f003:**
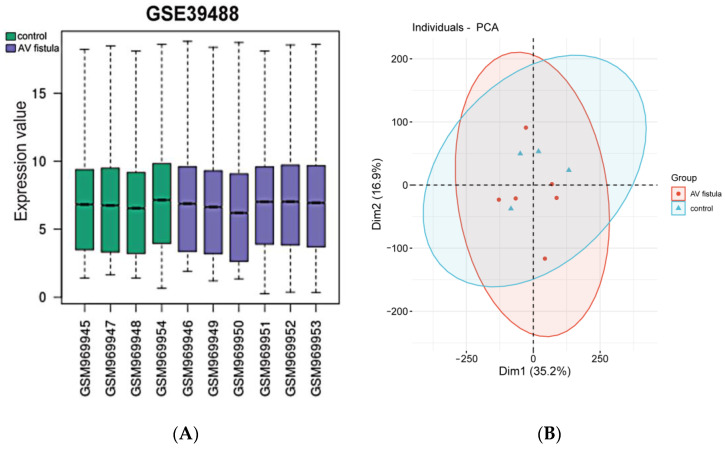
Consistency and repeatability analysis: (**A**) all samples from GSE39488 dataset were analyzed after log2 transformation, and the results suggest that mean values were approximately the same; (**B**) PCA model for the veins of stenotic AVF versus normal veins. PCA: principal component analysis.

**Figure 4 jpm-13-00207-f004:**
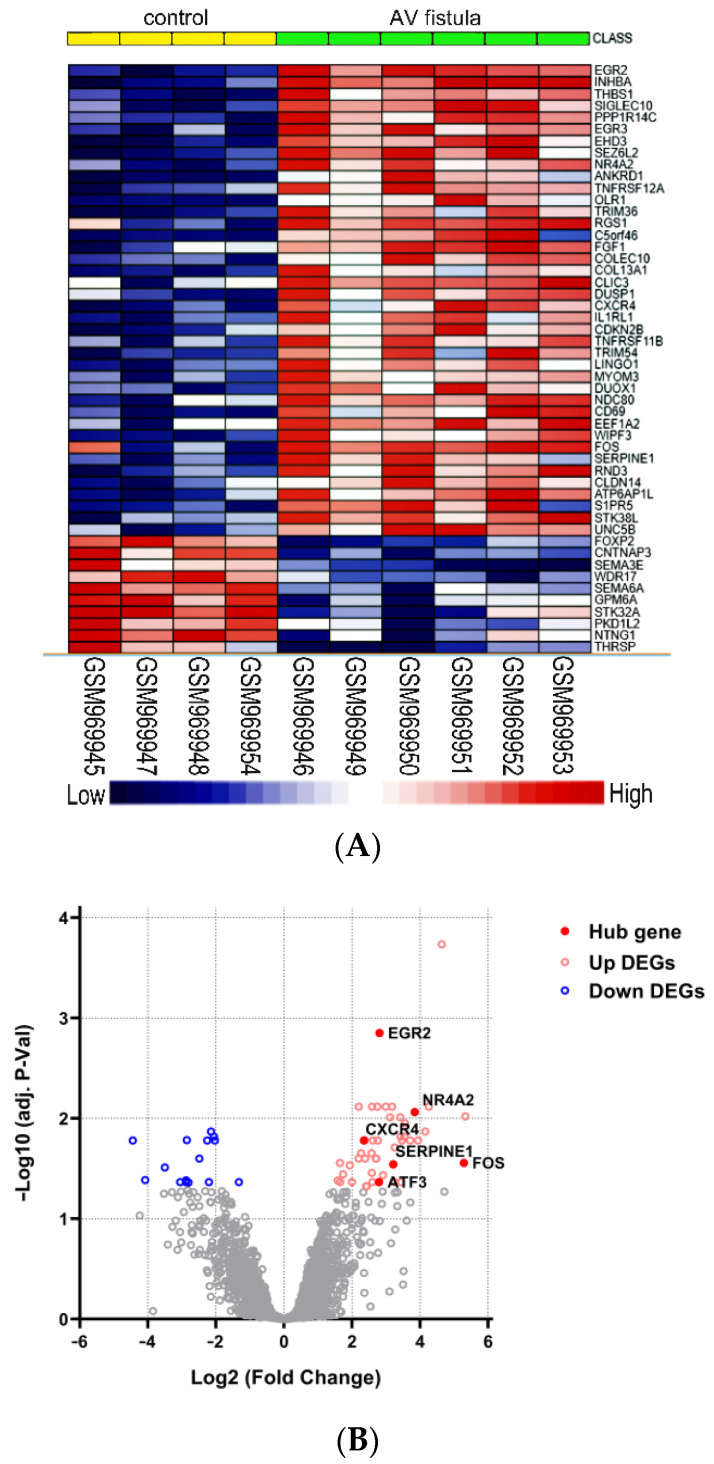
Identification of DEGs between control and AV fistula groups: (**A**) heatmap showing the top 50 DEGs between two groups; (**B**) volcano plot showing the DEGs between two groups after analysis using RStudio software. The red circles indicate upregulated genes, blue circles indicate downregulated genes, and the gray circles indicate genes which were not statistically different. DEG: differentially expressed gene.

**Figure 5 jpm-13-00207-f005:**
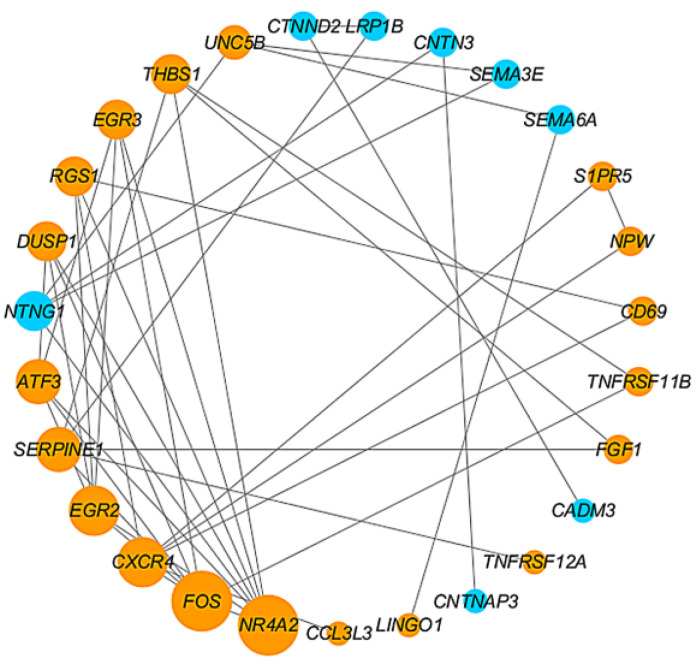
PPI network and hub genes. Size of the diameter represents the degree of the genes in the process of hub gene identification. The blue points indicate downregulated genes, and orange points indicate upregulated genes. PPI: protein–protein interaction.

**Figure 6 jpm-13-00207-f006:**
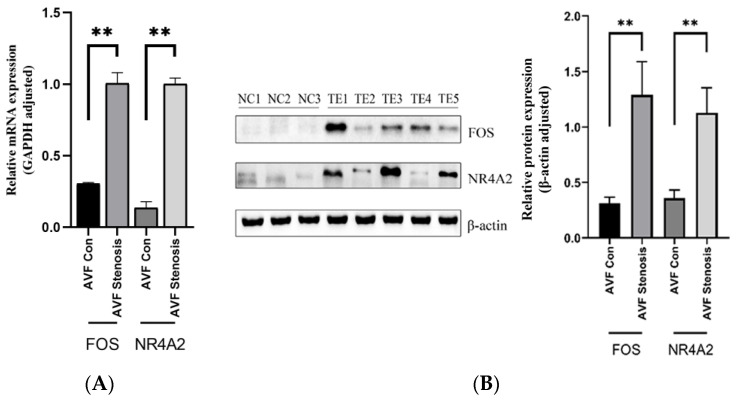
Expression levels of the mRNA and protein of *FOS* and *NR4A2*: (**A**) mRNA expression level; (**B**) protein expression level. Data analysis was performed using the one-way ANOVA test. ** *p* < 0.01.

**Figure 7 jpm-13-00207-f007:**
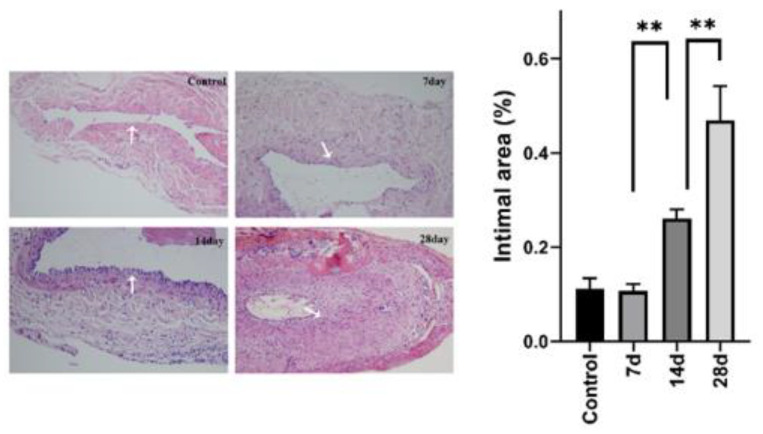
Histology results of the H&E-stained sections. Arrows indicate the thickened intimal of stenotic AVFs. Data analysis was performed using one-way ANOVA test. ** *p* < 0.01.

**Figure 8 jpm-13-00207-f008:**
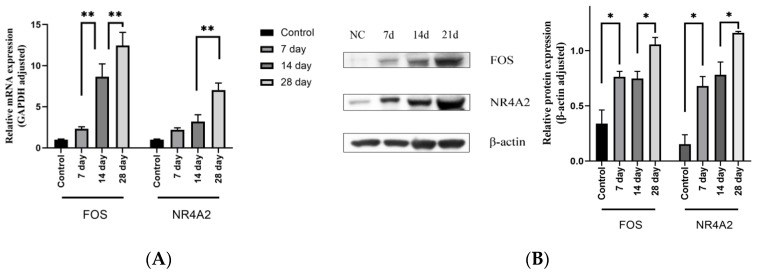
(**A**) mRNA expression level of *FOS* and *NR4A2* in rat samples. The expression level of mRNA gradually increased with the extension of time after surgery. (**B**) Protein expression level of *FOS* and *NR4A2*. Similar to the results for the mRNA, protein level gradually increased with the extension of time after surgery. Data analysis was performed using one-way ANOVA test. * *p* < 0.05 and ** *p* < 0.01.

**Table 1 jpm-13-00207-t001:** Clinical characteristics of patients.

	NC (*n* = 3)	Test (*n* = 5)	*p*-Value
Age (Y), Mean ± SD	56.9 (11.7)	58.1 (12.8)	0.865
Gender (Female, %)	0 (0%)	1 (20%)	0.408
Hypertension (Yes, %)	1 (33.3%)	3 (60%)	0.465
Diabetes mellitus (Yes, %)	2 (66.7%)	4 (80%)	0.673
Smoking (Yes, %)	3 (100%)	3 (60%)	0.206

**Table 2 jpm-13-00207-t002:** Primers used for RT-PCR.

Primer Name	Sequence (5′–3′)
Hsa-FOS-forward	GCGAGCTGTTCCCGTCAA
Hsa-FOS-reverse	CGTGGAAACCTGACGCAGAT
Hsa-NR4A2-forward	AGTCTGATCAGTGCCCTCGT
Hsa-NR4A2-reverse	TCGCCTGGAACCTGGAAT
Hsa-GAPDH-forward	GGTGAAGGTCGGAGTCAACG
Hsa-GAPDH-reverse	CAAAGTTGTCATGGATGHACC
Rno-FOS-forward	CCGGGGATAGCCTCTCTTACT
Rno-FOS-reverse	CCAGGTCCGTGCAGAAGTC
Rno-NR4A2-forward	GTTCAGGCGCAGTATGGGTC
Rno-NR4A2-reverse	CTCCCGAAGAGTGGTAACTGT
Rno-GAPDH-forward	GATGCTGGTGCTGAGTATGRCG
Rno-GAPDH-reverse	GTGGTGCAGGATGCATTGCTCTGA

## Data Availability

Original data used and/or analyzed during the current study are available from the corresponding author upon reasonable request.

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
