# Peer review of "Identification and Validation of Hub Genes in the Stenosis of Arteriovenous Fistula"

_jpm, 2023, doi:10.3390/jpm13020207_

Round 1
Reviewer 1 Report
Authors in their article analyzed samples of arteriovenous fistula (AVF) and identified potential candidate genes responsible for this condition. AVF and particularly its long-life patency, is very important for chronically dialyzed patients and thus knowledge of mechanisms leading to AVF stenosis are very important.
Study is designed correctly and technically satisfactory performed, however, its main limitation is the small size of samples analyzed making the conclusions not strong enough. Moreover, the dataset used (GSE39488) is based on microarray profiling which is nowadays quite old (and almost obsolete) method.
I have several comments:
1) Fig. 1: arrows indicating the direction of blood flow are not visible in the picture.
2) Fig. 2: presentation of PCA analysis is not much clear. I would prefer 2D plots over the 3D one.
3) Fig. 3: correct legend for panel B (volcano plot)
4) Fig. 4: presented results of GSEA analysis are not at all clear. Description how were these results obtained is neither in figure legend not in Methods section. Why MAPK pathway was analyzed at all? Statement: “The results of GSEA demonstrated that the top 3 pathways were MAPK signaling” is not documented at all. Based on Fig. 4, it seems to me that better candidates could be present in the dataset as well. Definitely, this section needs better support and documentation.
5) Fig. 6: expression of FOS and NR4A2 genes on protein level is quite heterogeneous in the samples. Samples used for this analysis should be described in more details (e.g. Were there differences in severity of stenosis?) as it is not clear what is the source of this variability. Moreover, quantitative analysis of WB is present, but there is no description in the Methods section how this was done. I would prefer to present the analysis for each individual sample as it seems that these is not (much) difference is some samples.
6) Fig. 7: two versions of Fig. 7 are present (and seems to be identical). How were calculated the values of intimal area presented in the quantitative analysis? There is no description in the Methods section… Did authors somehow standardize the section (its position within the sample) used for intimal area calculation (as such analysis should be significantly influenced by the site of analysis)?
7) Fig. 8: primers for PCR analysis of rat genes are not listed in Methods section.
8) Presented results of increased FOS and NR4A2 expression (Figs. 6, 8) may be caused just by the hypertrophy of intimal area (e.g. presence of many more cells compared to controls) without any change in their expression within each particular cell – did authors somehow checked for this possibility? Normalization to beta-actin (WB) or GAPDH (qPCR) will not resolve this question as the genes used for normalization are present in all cells within the sample.
9) English should be check thoroughly – there are many typos, some sentences are not clear and are hard to understand correctly. The Abstract should be entirely changed (too frequent usage of words like “however”, “moreover”…), a lot of typos are present, some sentences seem to be duplicated, the order of sentences is sometimes weird.
Reviewer 2 Report
Listed below are minor suggestions to the manuscript:
Figures:
Figure3B: Legend on the right can be more descriptive with longer annotations.
Figure3B: Also, the most significantly differentially expressed genes in the blue and orange color points can be labelled.
Figure3A: Font size of the gene names in the heatmap can be modified to make the annotations clearer. The names of the compared groups can be labelled in a better way.
Figure7: The formatting of the three figures in the figure7 needs modification. The first part of the figure on page7 needs to be consistent with the sizing of other figures.
Author Response
Reviewer 2
Thank you for the positive comments. We have modified our manuscript as requested.
(1) Figure3B: Legend on the right can be more descriptive with longer annotations. Figure3B: Also, the most significantly differentially expressed genes in the blue and orange color points can be labelled.
Response: Thank you for the comments. Firstly, figure 3B is not clear enough. Therefore, we redraw Figure 3B with the same data. In the new figure 3B, legends on the right were replaced as “Up DEGs” and “Down DEGs”, which were described in detail below. Meanwhile, new figure 3B was also attached as well. Secondly, in the new figure 3B, we have pointed out the hub genes.
(2) Figure3A: Font size of the gene names in the heatmap can be modified to make the annotations clearer. The names of the compared groups can be labelled in a better way.
Response: Thank you for the comments. Since we have listed the top 50 DEGs, it is very difficult to change the font size of gene names. The figures we upload are all 600dpi. Therefore, this may be solved by changing the size of figures when finalizing in the future. Moreover, we have modified the size of each sample. And we added the name of each group (“control” and “AV fistula”) above Fig. 3A.
(3) Figure7: The formatting of the three figures in the figure7 needs modification. The first part of the figure on page7 needs to be consistent with the sizing of other figures.
Response: Thank you for the comment. Because of our negligence, we made a small mistake. Fig. 7 on page 7 and page 8 are duplicate, so we deleted one version of Fig.7.
Reviewer 3 Report
The article by Li et al. describes their research on the steonosis of arteriovenous fistula. The concept is well-described and the results and conclusions are sound. Although the number of samples is quite low, it is understandable since the access to such primary material is not easy.
I have minor comments regarding the manuscript:
Fig. 3 A - please include a label for the "class" row at the top - what does red and what does blue mean here? It would also be good to change the colors since they are already used as a measure of expression changes.
Fig. 4 - the authors present only a single GSEA plot, while they mention the upregulation of NOD-like receptor signaling and cytokine receptor signaling. Please provide appropriate results in this figure. Please include a NES, p-value and FDR.
Fig. 7 seems to be duplicated partially on the previous page as well? Please correct this layout problem.
Overall the readability of the figures could be improved by adjusting the relative panel sizes and the font sizes so that they are more or less the same. Especially fig 6, fig 8.
Even though the statistical significance is denoted with an asterisk, there is a lack of description of the statistical test used in the figure legend. Please mention these whenever appropriate tests were used.
Author Response
Reviewer 3
The article by Li et al. describes their research on the stenosis of arteriovenous fistula. The concept is well-described and the results and conclusions are sound. Although the number of samples is quite low, it is understandable since the access to such primary material is not easy.
Response: Thank you for the positive comment. Those comments are all valuable and helpful for improving our manuscript, as well as guiding our future research. Following your suggestions, we have modified our manuscript point by point.
(1) Fig. 3A: please include a label for the "class" row at the top - what does red and what does blue mean here? It would also be good to change the colors since they are already used as a measure of expression changes.
Response: Thank you for the comments. We have added the name of each group above the “class” row (“NC” and “AV fistula”). Meanwhile, we have also changed the colors of the “class” row.
(2) Fig. 4: The authors present only a single GSEA plot, while they mention the upregulation of NOD-like receptor signaling and cytokine receptor signaling. Please provide appropriate results in this figure. Please include a NES, p-value and FDR.
Response: Thank you for the comment. We have considered your comments carefully. We found that in our manuscript, the results of western blot and RT-PCR may have little relationship with the result of GSEA. FOS, one of the hub genes, could be involved in MAPK signaling pathway. However, this is not our main research content. In summary, we deleted the content about GSEA. Meanwhile, we rewrote other sections in our manuscript including GSEA.
(3) Fig. 7 seems to be duplicated partially on the previous page as well? Please correct this layout problem.
Response: Thank you for the comments. We made this mistake due to our negligence. We have corrected this layout as requested.
(4) Overall, the readability of the figures could be improved by adjusting the relative panel sizes and the font sizes so that they are more or less the same. Especially fig 6, fig 8.
Response: Thank you for the comments. This suggestion is very helpful. We have adjusted the panel size and the font size of Fig. 6 and Fig. 8 as requested. New figures have also been added in our manuscript.
(5) Even though the statistical significance is denoted with an asterisk, there is a lack of description of the statistical test used in the figure legend. Please mention these whenever appropriate tests were used.
Response: Thank you for the comments. We have added the statistical test in each figure legend. We added following sentence “Data analysis was performed by one-way ANOVA test.”
Round 2
Reviewer 1 Report
I would like to thank authors for thorough revision of the paper and addressing all my points. The paper is now much improved and I have just a couple of minor points:
1) use “Hsa” instead of “Has” in Table 2 (list of primers)
2) the labels in Fig. 3, panel B are difficult to read; Y axis label looks weird; there is a lot of empty space – be sure that in the final version of the Figure has correct resolution and format.
3) in Fig. 6 legend *, **, and *** are mentioned, however, only ** is actually used in the Figure.